# Combination Therapy with a TLR7 Agonist and a BRD4 Inhibitor Suppresses Tumor Growth via Enhanced Immunomodulation

**DOI:** 10.3390/ijms25010663

**Published:** 2024-01-04

**Authors:** Yong-Si Liu, Jia-Xin Wang, Guang-Yi Jin, Ming-Hao Hu, Xiao-Dong Wang

**Affiliations:** Nation-Regional Engineering Lab for Synthetic Biology of Medicine, International Cancer Center, School of Pharmacy, Shenzhen University Medical School, Shenzhen 518060, China; yongsilau@163.com (Y.-S.L.); wangjiaxin3742@163.com (J.-X.W.); gyjin@szu.edu.cn (G.-Y.J.)

**Keywords:** TLR7, BRD4, antitumor, immunity, TAM

## Abstract

JQ-1 is a typical BRD4 inhibitor with the ability to directly fight tumor cells and evoke antitumor immunity via reducing the expression of PD-L1. However, problems arise with the development of JQ-1 in clinical trials, such as marked lymphoid and hematopoietic toxicity, leading to the investigation of combination therapy. SZU-101 is a TLR7 agonist designed and synthesized by our group with potent immunostimulatory activity. Therefore, we hypothesized that combination therapy of SZU-101 and JQ-1 would target innate immunity and adaptive immunity simultaneously, to achieve a better antitumor efficacy than monotherapy. In this study, the repressive effects of the combination administration on tumor growth and metastasis were demonstrated in both murine breast cancer and melanoma models. In 4T1 tumor-bearing mice, i.t. treatment with SZU-101 in combination with i.p. treatment with JQ-1 suppressed the growth of tumors at both injected and uninjected sites. Combination therapy increased M1/M2 ratio in TAMs, decreased PD-L1 expression and promoted the recruitment of activated CD8^+^ T cells in the TME. In summary, the improved therapeutic efficacy of the novel combination therapy appears to be feasible for the treatment of a diversity of cancers.

## 1. Introduction

Toll-like receptors (TLRs) are able to recognize pathogen-associated molecular patterns (PAMPs), thereby acting as the major bridge between innate immunity and adaptive immunity [1]. There are more than 10 types of TLRs, and TLR7 is located intracellularly on the membranes of endosomes in a variety of immune cells like macrophages, DCs, T cells and B cells [2]. TLR7 recognizes a series of single-stranded RNA sequences and synthetic agonists. TLR7 agonists demonstrate potent immunostimulatory activity to fight against many types of tumors through the induction of tumor-specific immune responses [3,4]. Imiquimod, a typical small molecule agonist, was approved in 2004 by the FDA for the treatment of superficial basal cell carcinoma [5]. Our laboratory has been committed to the design and synthesis of small-molecule TLR7 agonists, with SZU-101 as a representative compound. SZU-101 has been displayed as a successful adjuvant in the construction of tumor vaccines targeting gastric cancer and breast cancer [6,7]. Its uses in the form of chemical conjugation with targeted drugs, such as ibrutinib, have also been reported [8,9].

The BET (bromodomain and extra-terminal domain) family of proteins were first reported as epigenetic regulators in inflammatory diseases and now are verified to have necessary roles in tumorigenesis through the remodeling of aberrant chromatin and the mediation of gene transcription [10,11]. BRD4 (bromodomain-containing protein 4) is a typical member of the BET proteins with the most extensive investigation, and small-molecule BRD4 inhibitors with high potency and specificity, such as JQ-1, have been developed [12]. The antitumor efficacy of JQ-1 has always been attributed mainly to the ability to depress the transcription of c-Myc [13]. Recently, JQ-1 has been shown to restore the antitumor immunity by reducing the expression of another direct BRD4-mediated gene, CD274 (encoding PD-L1). PD-L1 (programmed cell death-ligand 1) is a common ligand of the immune checkpoint PD-1 (programmed cell death-1). The binding of PD-1 and PD-L1 creates an environment of immune escape via the suppression of T cell activity [14]. However, problems arise with the development of JQ-1 in clinical trials, such as the remarkable toxicity towards hematopoietic and lymphoid tissues [15]. In order to overcome the defects, some novel approaches are being studied, like combination administration of JQ-1 with other drugs [16,17].

The main targets of TLR agonists and PD-1 blockades are innate immune cells and adaptive immune cells, respectively. It is reported that a TLR7 agonist or a TLR9 agonist (i.t. treatment) enhances the efficacy of anti-PD-1 antibody (i.p. treatment) through the activation of CD8^+^ T cells [3]. Therefore, in this study, we hypothesized that combination therapy of SZU-101 and JQ-1 would target innate immunity and adaptive immunity simultaneously to achieve a better antitumor efficacy than SZU-101 or JQ-1 monotherapy. First, the immunostimulatory activity of SZU-101 and JQ-1 was decided in vitro. Then, we demonstrated that i.t. treatment with SZU-101 in combination with i.p. treatment with JQ-1 suppressed the growth of tumors at both injected and uninjected sites. Combination therapy increased M1/M2 ratio in TAMs (tumor-associated macrophages), decreased the expression of PD-L1 and promoted the recruitment of activated CD8^+^ T cells in the TME (tumor microenvironment). Finally, the antimetastatic effects of combination therapy were further validated in two murine tumor models. Here, the 4T1 breast cancer model and the B16 melanoma model were applied because the antitumor effects of TLR7 agonists or BRD4 inhibitors were decided previously in these two types of cancers [7,8,9]. Taken together, SZU-101 appeared to represent an efficient way of improving the antitumor potency of JQ-1.

## 2. Results

### 2.1. SZU-101 Stimulated TLR7 Signaling to Prime Immune Responses In Vitro

SZU-101 was the novel TLR7 agonist previously designed and synthesized by our group [8] (Appendix A). TLR7-NF-κB and TLR8-NF-κB reporter systems in HEK-293 cells were constructed and treated with SZU-101, Imiquimod (a typical TLR7 agonist) and R848 (a typical TLR7/8 agonist). SZU-101 and R848 exhibited similar effects on TLR7 stimulation, much more potent than Imiquimod, while SZU-101 and Imiquimod exhibited no effects on TLR8 stimulation (Appendix A). Thus, SZU-101 was an agonist targeting TLR7 with selectivity and specificity. Then, mouse BMDCs and spleen lymphocytes were exposed to SZU-101 overnight to examine the ability on the production of cytokines, including IL-12, TNF-α (T_H_-1-biassed) and IL-6 (T_H_-2-biassed). SZU-101 displayed remarkable effects on the increase in cytokine release in a dose-dependent manner (from 1.5 to 15 μM) (Appendix A). These in vitro data indicated that SZU-101 was effective on the stimulation of TLR7 and the following immune responses.

### 2.2. JQ-1 Displayed Growth Inhibition on Tumor Cells In Vitro

JQ-1 inhibited the growth of mouse 4T1 breast cancer cells in dose-dependent (from 3.125 to 100 μM) and time-dependent (from 1 to 3 day) manners, determined by CCK-8 assay (Appendix A). Cell apoptosis was assessed by Annexin V-FITC/PI double staining, where JQ-1 led to a notable occurrence of early and late apoptotic populations in 4T1 cells (Appendix A). Key proteins related to apoptosis and autophagy were also detected by Western blot. Cleavage of Caspase-3 and PARP, a transformation of LC3-I to LC3-II and a decline in P62 were all be recorded in the cells treated with high concentrations of JQ-1 (≥10 μM) (Appendix A). The expression of PD-L1 (a direct BRD4-mediated element) was examined by Western blot and flow cytometry. Low concentrations of JQ-1 (≤1 μM) decreased PD-L1 expression both in the whole cell and on the surface of 4T1 cells (Appendix A). Similar results of JQ-1 in mouse B16 melanoma cells were reported previously by our group [8].

### 2.3. Combination Administration of SZU-101 and JQ-1 Inhibited Tumor Growth at Both Injected and Uninjected Sites

In order to determine the dose of SZU-101 for in vivo administration, Balb/c mice were implanted with 4T1 cells on Day 0 on the right flank, and intratumorally (i.t.) treated with SZU-101 for successive 5 days (from Day 7 to Day 11). Measures of 10 mg/kg and 30 mg/kg SZU-101 displayed similar antitumor effects, while no significant difference was detected between negative control and 3 mg/kg drug treatment (Figure 1A). Since the dose of 30 mg/kg was high enough to arouse the side effects, such as cytokine storm, 10 mg/kg was selected for the following experiments.

Next, the optimal drug regimen was decided by comparing three types of schedules of SZU-101 (10 mg/kg) administration: (i) discontinuous i.t. treatment (intratumoral administration for 5 times, from Day 7 to Day 19), (ii) continuous i.p. treatment (intraperitoneal administration for 5 times, from Day 7 to Day 11), (iii) continuous i.t. treatment (intratumoral administration for 5 times, from Day 7 to Day 11). Only continuous i.t. treatment displayed potent antitumor effects, compared to negative control (Figure 1B).

To evaluate the suppressive effects on tumor growth by the combination of SZU-101 and JQ-1, Balb/c mice were implanted with 4T1 cells on both flanks and treated with SZU-101 (i.t.) and JQ-1 (i.p.). The treatment schedule of JQ-1 was settled according to the previous reports [13,14]. Monotherapy with SZU-101 or JQ-1 inhibited tumor growth at both injected and uninjected sites. SZU-101 enhanced the suppressive efficacy of JQ-1 when two drugs were used in combination (Figure 1C and Appendix A). Tumor weights were evaluated at the end of the experiments with similar results as tumor growth curves (Figure 1D). SZU-101 in combination with JQ-1 prolonged the survival rate to 3 of 8 mice alive within 70-day observation (Figure 1E). C57BL/6J mice were implanted with B16 cells and treated with SZU-101 and JQ-1, and similar inhibitory effects of combination therapy on tumor growth were demonstrated on both injected and uninjected sites (Appendix A).

### 2.4. Combination Administration of SZU-101 and JQ-1 Increased M1/M2 Ratio in TAMs

Tumor-infiltrating cells were harvested on Day 12 (24 h after the final SZU-101 treatment) and Day 21 (the end of the experiment) in order to investigate the promotion of the antigen-presenting ability by SZU-101 of the immune cells in the TME. TAMs were characterized as a CD45^+^CD11b^+^F4/80^+^ subset, and two phenotypes of TAMs, M1-like (CD206^−^) and M2-like (CD206^+^) macrophages, were examined by flow cytometry (Appendix A). SZU-101 monotherapy and combination therapy increased the ratio of M1 to M2 macrophages (M1/M2 ratio) at the injected site on both Day 12 and Day 21. No significant difference in M1/M2 ratio at the uninjected site could be detected between drug treatment and control groups (Figure 2A,B). To understand the kinetics of M1 and M2 populations after SZU-101 treatment, the expressions of CD206 and MHC class II were examined at the injected site, further clarifying M1 and M2 as CD206^−^MHC class II^+^ and CD206^+^MHC class II^−^ subsets. CD206 expression was downregulated by SZU-101 monotherapy and combination therapy on Day 12 and Day 21, while MHC class II expression was upregulated by SZU-101 monotherapy and combination therapy only on Day 21 (Figure 2C,D).

### 2.5. Combination Administration of SZU-101 and JQ-1 Suppressed PD-L1 Expression in Tumor Cells

Tumor cells (CD45^−^ subset) were collected to investigate whether JQ-1 could inhibit immunosuppressive factors, with the expression of PD-L1 determined by flow cytometry on Day 21. Similar consequences of PD-L1 inhibition were obtained for either JQ-1 monotherapy or combination therapy at both injected and uninjected sites (Figure 3A). The PD-L1 expression was also confirmed by immunohistochemical staining, where JQ-1 monotherapy and combination therapy significantly decreased PD-L1 expression compared to negative control and SZU-101 monotherapy (Figure 3B).

### 2.6. Combination Administration of SZU-101 and JQ-1 Increased CD8^+^ T Cells in Spleens and TILs

CD8^+^ T cells and activated CD8^+^ (CD8^+^IFNγ^+^) T cells in spleens and tumors were determined by flow cytometry in order to investigate whether the promotion of TAMs and the inhibition of immunosuppressive factors could induce the recruitment and the activation of tumor-specific T cells in tumors (Appendix A). Combination administration of SZU-101 and JQ-1 increased the numbers of CD45^+^CD3^+^CD8^+^ and CD8^+^IFNγ^+^ T cells in spleens (Figure 4A). LDH assay was applied to analyze CTL activity, with spleen lymphocytes from 4T1 tumor-bearing mice and 4T1 cells being employed as effector cells and target cells, respectively. Combination administration exhibited greater CTL lytic activity to 4T1 cells than any other groups (Figure 4B). In tumors at both injected and uninjected sites, the numbers of total CD8^+^ and CD8^+^IFNγ^+^ T cells were significantly elevated by combination therapy (Figure 4C). Increased CD8^+^ T cell infiltration in tumors was also confirmed by immunohistochemical staining (Figure 4D).

### 2.7. Depletion of CD8^+^ Cells Abrogated the Antitumor Effects of Combination Administration of SZU-101 and JQ-1

Depletion of CD8^+^ cells by anti-CD8 mAb was performed to investigate the relationship between tumor growth inhibition and tumor-specific CD8^+^ T cells activated by combination therapy, especially the distant tumors. On Day-1, 3, 7, 11, 14 and 18, mice treated with SZU-101 and JQ-1 were further intraperitoneally injected with anti-CD8 or isotype control antibody (Figure 5A). Tumor volumes and tumor weights were decided accordingly. With the depletion of CD8^+^ cells, no difference in tumor volumes or tumor weights could be detected between negative control and combination therapy at either injected or uninjected sites (Figure 5B,C). These findings suggested that CD8^+^ T cells induced by combination administration could suppress the metastatic tumor growth.

### 2.8. Combination Administration of SZU-101 and JQ-1 Inhibited Tumor Metastasis

To investigate whether combination therapy could suppress tumor metastasis, Balb/c mice were intravenously injected through the tail vein with 4T1 cells on Day 0, and i.p. treated with SZU-101 and JQ-1 (Figure 6A). Numbers of lung nodules were counted on Day 21. JQ-1 inhibited lung metastasis effectively, and SZU-101 enhanced the suppressive efficacy of JQ-1 when two drugs were used in combination (Figure 6B and Appendix A). Combination therapy also significantly prolonged the survival time of the mice (Figure 6C). CD8^+^ T cells and activated CD8^+^ (CD8^+^IFNγ^+^) T cells in spleens were determined by flow cytometry. Combination therapy elevated the levels of total CD8^+^ and CD8^+^IFNγ^+^ T cells compared to negative control (Figure 6D). CTL activity was determined by LDH assay, and combination administration exhibited the greatest lytic activity to 4T1 cells (Figure 6E). Inhibition of tumor metastasis was also verified in a murine melanoma model, where B16 cells were intravenously injected into C57BL/6J mice through the tail veins, and combination therapy potently inhibited the numbers of lung nodules (Appendix A).

## 3. Discussion

Over the past decade, immunotherapy has displayed tremendous potential in many malignancies so as to overcome the recurrence and metastases of tumors. Immune checkpoints, such as PD-1 and its ligand PD-L1, are the attractive targets in cancer immunotherapy [18]. Antagonistic antibodies of PD-1 or PD-L1 are put into use for the treatment of cancers with sustained tumor regression and acceptable long-term safety [19,20]. However, intrinsic and acquired resistance of the antibodies requires other strategies to restrain immune checkpoints. Specific small-molecule inhibitors could be promising candidate drugs, possessing the potential to both suppress PD-L1 expression and inhibit oncogenic signaling pathways, like JQ-1, the typical BRD4 inhibitor [21]. Since JQ-1 mainly targets adaptive immune cells, we hypothesized that our TLR7 agonist, SZU-101, could improve its antitumor effects by activating innate immune cells simultaneously.

SZU-101 was synthesized and reported by our group previously, and here, we systematically investigated its immunostimulatory activity. SZU-101 was a potent agonist targeting TLR7 specifically, but not TLR8, and induced the production of both T_H_-1-biassed and T_H_-2-biassed cytokines in mouse BMDCs and spleen lymphocytes (Appendix A). BRD4 inhibitors downregulate the expression of a series of transcription factors (c-Myc, PD-L1, etc.) via the disruption of the binding of BRD4 protein to acetylated lysine residues of chromatin [22]. JQ-1, a BRD4 inhibitor, has been verified to repress tumor growth by inducing direct cell death and indirect immune activation in multiple human cancer cells [23,24]. However, the effects of JQ-1 on mouse 4T1 breast cancer cells are seldomly discussed. We demonstrated that JQ-1 inhibited the viability of 4T1 cells in both dose-dependent and time-dependent manners, through the pathways of apoptosis and autophagy, especially in the cells treated with higher concentrations of JQ-1 (≥10 μM). PD-L1 inhibition could be detected in the cells treated with much lower concentrations of JQ-1 (≤1 μM) (Appendix A). The above in vitro results inferred that JQ-1 concurrently suppressed tumor growth and restored antitumor immunity, and PD-L1 inhibition could be easier to reach than other effects.

TLR agonists are given either systemically or locally in clinical trials [25], and SZU-101 is given either continuously or discontinuously in different types of tumors [8,9]. Systemic administration of TLR agonists is reported to cause side effects like flu-like symptoms and lymphopenia [26]. Immune unresponsiveness could be triggered when TLR agonists are administrated with a continuous and systemic schedule [27]. Therefore, we compared three schedules of drug administration, and it turned out that a continuous i.t. schedule of SZU-101 at 10 mg/kg (from Day 7 to Day 11) was the most effective one (Figure 1B). JQ-1 was delivered with a discontinuous i.p. schedule at 50 mg/kg (from Day 7 to Day 18) as reported before. From the results of tumor volumes and tumor weights, combination therapy displayed more potent antitumor effects than monotherapy with SZU-101 or JQ-1 at both the injected and uninjected sites (Figure 1C,D). The therapeutic efficacy of combination therapy was also mirrored by the distinct extension of the survival curve of 4T1-bearing mice (Figure 1E). Additionally, we reported the antitumor effects of a chemical conjugation of SZU-101 and JQ-1 (SZU-119), where JQ-1 alone was included as a comparison group and given to the mice with the same dose and schedule as SZU-119 (10 mg/kg, continuous i.t. treatment for 5 times) [8]. However, JQ-1 alone could achieve stronger antitumor effects without apparent toxicity by being administrated with a higher dose (50 mg/kg) and a discontinuous i.p. treatment 5 times. In this work, a strategy of combination therapy with SZU-101 and JQ-1 with different drug administration plans was verified with more potent effects on tumor growth than the chemical conjugation SZU-119.

TAMs are commonly classified into two phenotypes, M1-like macrophages (classically activated state) and M2-like macrophages (alternatively activated state). TAMs derived from peripheral blood monocytes can be recruited to the TME and turned into either M1 or M2 population in response to different stimuli produced by tumors [28,29]. M1 macrophages are involved in antitumor effects by producing nitric oxide, reactive oxygen species and proinflammatory cytokines [30]. MHC class II molecules of M1 macrophages facilitate the antigen-presenting capacity and shape the adaptive immune system. In contrast, M2 macrophages are considered to promote tumor initiation and progression via immunosuppressive cytokines like IL-10 and TGF-β [31]. CD206, CD204 and CD163 are representative surface proteins of M2 macrophages [32]. We demonstrated that SZU-101 elevated the M1/M2 ratio in the injected site on both Day 12 and Day 21, and combination therapy exerted similar effects to SZU-101 monotherapy (Figure 2A,B). These findings were consistent with the previous report that TLR7 agonists can remodel M2 phenotype into M1 phenotype [33]. Kinetics of M1 and M2 populations was further determined, as shown in Figure 2C,D. A reduction in the M2 population occurred on Day 12, one day after the final treatment of SZU-101, but expansion of the M1 population could only be detected on Day 21, a later time point. JQ-1 is proven to depress the expression of PD-L1 on tumor cells and immune cells to promote antitumor immunity [14]. Here, a decrease in PD-L1 expression was observed in the TME of both injected and uninjected tumors treated with either JQ-1 alone or in combination with SZU-101, illustrated by flow cytometry and histologic analysis (Figure 3).

CD8^+^ T cells are the main effector cells in cellular immune responses with surface inhibitory receptor PD-1. Blocking of the binding of PD-1 and PD-L1 can conserve the cytolytic function [34]. Polarization of TAMs is of great importance in the recruitment of CD8^+^ T cells to TME to directly mediate cytotoxicity [35]. In this project, as shown in Figure 4A,B, the numbers and activity of CD8^+^ T cells were first determined in spleens. Combination therapy elevated the numbers of both total CD8^+^ T cells and CD8^+^IFNγ^+^ T cells, along with the CTL lytic ability. Results of flow cytometry and histologic analysis indicated that combination therapy boosted the numbers of activated CD8^+^ T cells at both injected and uninjected sites, inferring that an expansion of tumor-reactive CD8^+^ T cells occurred locally and systemically (Figure 4C,D). Accompanied by the treatment of SZU-101 and JQ-1, CD8^+^ cells were silenced by monoclonal antibodies to further study the relationship between cytotoxic T cell immune responses and antitumor outcomes. As expected, depletion of CD8^+^ T cells reversed the suppression of tumor volumes and tumor weights by combination therapy at both local and abscopal sites (Figure 5).

Ninety percent of the deaths in tumor patients are attributed to metastasis, and lungs are among the most frequent sites of tumor metastasis owing to their immunosuppressive microenvironment [36,37]. TLR7 agonists are proven to activate immune responses in several types of pulmonary metastatic cancers by the induction of oligoclonal CD8^+^ T cells [3]. Antimetastatic effects of SZU-101 in combination with JQ-1 were verified in the syngeneic tumor mouse model of 4T1 cells. As shown in Figure 6, combination therapy significantly decreased the numbers of lung nodules and increased the survival time of the mice. In spleens, numbers of total CD8^+^ T cells and activated CD8^+^ T cells were elevated, while cytotoxic T cell activity against the tumor cells was also strengthened, compared to negative control. Moreover, antitumor efficacy was monitored in immunocompetent C57BL/6J mice bearing the syngeneic B16 melanoma cells in both the tumor growth model and the tumor metastasis model. SZU-101 was given with a continuous i.t. schedule, and JQ-1 was given with a discontinuous i.p. schedule. After the treatment of SZU-101 together with JQ-1, the reduction of tumor volumes at injected and uninjected sites was more distinct than the application of SZU-101 or JQ-1 alone (Appendix A). Results of Appendix A indicated that SZU-101 or JQ-1 alone could not reduce the numbers of lung nodules efficiently, but combination therapy exerted sufficient inhibitory effects on tumor metastasis. Therefore, cooperation involving two mechanisms on the innate immunity by a TLR7 agonist and the adaptive immunity by a BRD4 inhibitor was required for the optimal suppressive effects of aggressive tumors. Of course, the immunomodulatory effects of combination therapy of SZU-101 and JQ-1 should be explained in more detail in the future, such as the ability of antigen uptake of macrophages and the frequency of TCR clones of T cells. In addition, the antitumor effects and mechanisms could also be verified in other types of cancers.

## 4. Materials and Methods

### 4.1. Animals, Reagents and Cell Lines

Female Balb/c mice and C57BL/6J mice (6 to 8 weeks old) were purchased from Medical Laboratory Animal Center (Guangzhou, China). JQ-1 was purchased from Selleck (Shanghai, China), and SZU-101 was synthesized by our group [8]. Mouse 4T1 breast cancer cells and mouse B16 melanoma cells (ATCC, Manassas, VA, USA) were cultured at 37 °C in a humidified atmosphere with 5% CO_2_ in DMEM medium (10% fetal bovine serum, 100 U/mL penicillin and 100 μg/mL streptomycin). Cells were harvested from the culture during the exponential growth phase.

### 4.2. Tumor Implantation Models

Each mouse was subcutaneously inoculated in the right flank or both flanks with 2 × 10^5^ 4T1 or B16 cells. When the diameters of the tumors were about 2–4 mm (on Day 7 after inoculation), treatment with compounds was ready to start. A total of 10 mg/kg SZU-101 (intratumoral injection to the right flank) for successive 5 days, or 50 mg/kg JQ-1 (intraperitoneal injection) three times per week was given to the mice. Detailed experimental protocols were shown in Figure 1. Tumor volumes were calculated by the following formula: volume (mm^3^) = ([width]^2^ × length)/2. On Day 21, the mice were sacrificed, while the tumors were harvested and weighed. Long-term survival was also evaluated until the mice died naturally or the tumor diameters reached 20 mm.

### 4.3. Lung Metastasis Models

Each mouse was intravenously injected through the tail vein with 5 × 10^4^ 4T1 or B16 cells. Treatment with compounds was initiated on Day 7 after injection. Briefly, mice received the treatment with 10 mg/kg SZU-101 (intraperitoneal injection) for 5 successive days, or 50 mg/kg JQ-1 (intraperitoneal injection) three times per week. Detailed experimental protocols were shown in Figure 6. On Day 21, the mice were sacrificed, and the lungs were harvested and fixed in Bouin’s solution to count the numbers of lung surface metastatic nodules. Long-term survival was also evaluated until the mice died naturally.

### 4.4. Flow Cytometric Analysis

Spleens were homogenized, while tumors were homogenized and incubated with collagenase D (1 mg/mL) and DNase I (0.05 mg/mL) (Sigma-Aldrich, St. Louis, MO, USA) at 37 °C for 1 h. Single cell suspensions of spleens and tumors were stained with the antibodies to surface markers at 4 °C for 30 min, and intracellular IFN-γ staining was performed by Fixation/Permeabilization Solution kit (eBioscience, San Diego, CA, USA). The fluorescence was analyzed by FACSCalibur flow cytometry (BD Biosciences, San Jose, CA, USA). The antibodies were described in Appendix A.

### 4.5. Histologic Analysis

Tumors and lungs were fixed with 4% paraformaldehyde, dehydrated and embedded in paraffin. After antigen retrieval, sections (5 μm) were incubated with 3% H_2_O_2_ to remove endogenous peroxidase, and then incubated with blocking buffer for 1 h. The primary antibodies (rabbit anti-mouse CD8, ab237723, 1:200 dilution; rabbit anti-mouse PD-L1, ab233482, 1:100 dilution) (Abcam, Cambridge, UK) were incubated at 4 °C overnight, while the secondary antibody (HRP goat anti-rabbit IgG, ab205718, 1:2000 dilution) was incubated at room temperature for 30 min. Lastly, the slides were stained with DAB and counterstained with hematoxylin.

### 4.6. CTL Analysis

Spleens of the mice were harvested, and lymphocytes functioned as the effectors were separated by Mouse Lymphocyte Separation Medium (Dakewe, Beijing, China), according to the supplier’s manual. The 4T1 tumor cells functioning as the targets were co-cultured with lymphocytes at the ratio of cell number of 1:50 for 4 h. Cytotoxicity of CTLs was decided by LDH method using Non-Radioactive Cytotoxicity Assay (Promega, Madison, WI, USA), according to the supplier’s manual.

### 4.7. CD8^+^ Cell Depletion In Vivo

Mouse anti-CD8 mAb (clone 2.43) and comparable isotype control (clone LTF-2) were purchased from BioXcell (West Lebanon, NH, USA). Anti-CD8 or isotype control antibody was intraperitoneally administrated to the mice at a dose of 200 μg on Day-1, 3, 7, 11, 14 and 18, as displayed in Figure 5. More than 90% depletion of CD8^+^ cells were verified by flow cytometry.

### 4.8. Statistical Analysis

Data were expressed as mean ± SE for the indicated number of independently performed experiments. In dot plots, each dot represented a tumor or a spleen from an individual mouse. One-way ANOVA with Tukey’s post hoc test was used to compare multiple groups. Two-way ANOVA with Bonferroni post hoc test was used in the experiment of the tumor volumes collected over all time points. Log rank (Mantel-Cox) test was used to test for the significant difference between survival curves. GraphPad Prism software vesion 8.0.1 (San Diego, CA, USA) was used to carry out these analyses. A value of *p* < 0.05 was considered statistically significant.

## 5. Conclusions

In conclusion, our study displayed that SZU-101 (a TLR7 agonist synthesized by our group) and JQ-1 (a well-defined BRD4 inhibitor) could be administrated in combination to achieve better repressive effects on tumor growth and metastasis in both murine breast cancer and melanoma models. In 4T1 tumor-bearing mice, combination therapy produced systemic antitumor results more potently than the monotherapy. Some immunologic parameters associated with the efficacy were fully discussed as follows. First, M2-like macrophages were remodeled to M1-like macrophages at the injected site. Then, PD-L1 inhibition and CD8^+^ T cell activation were ascertained at both injected and uninjected sites. The importance of the CD8^+^ T cells with a tumoricidal function was further verified by the depletion with monoclonal antibody (Figure 7). Taken together, the improved therapeutic efficacy of the novel combination therapy appears to be feasible for the treatment of a diversity of cancers in the further clinical trials with this regimen.

## Figures and Tables

**Figure 1 ijms-25-00663-f001:**
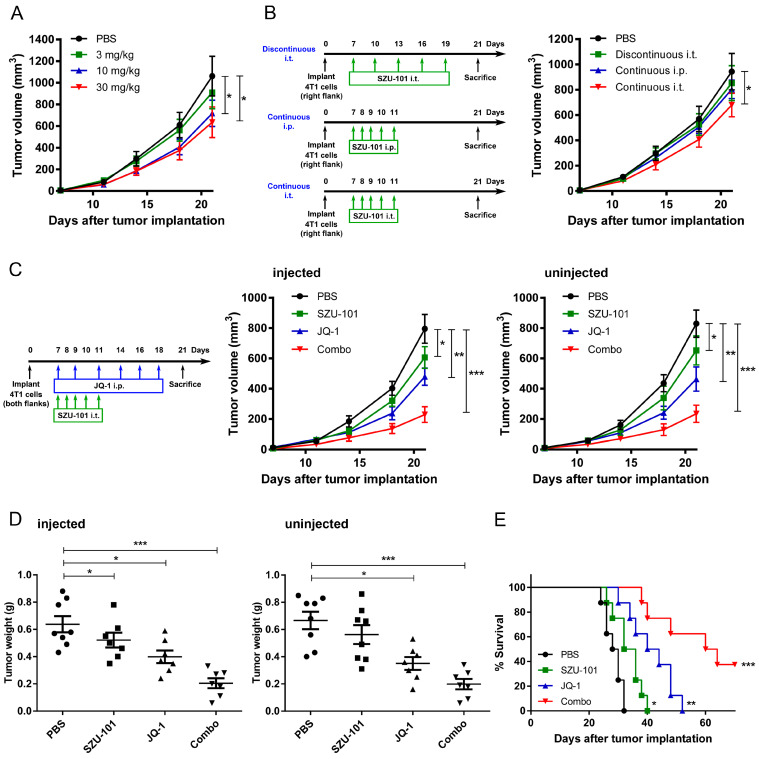
Combination administration of SZU-101 and JQ-1 inhibited tumor growth at both injected and uninjected sites. (**A**) Dose optimization studies of i.t. administration of SZU-101. Balb/c mice (*n* = 5–7/group) were implanted with 2 × 10^5^ 4T1 cells in the right flank on Day 0, and i.t. treated with 3, 10 or 30 mg/kg SZU-101 for successive 5 days from Day 7 to Day 11. (**B**) Schedule optimization studies of administration of 10 mg/kg SZU-101. Balb/c mice (*n* = 5–7/group) were implanted with 2 × 10^5^ 4T1 cells in the right flank on Day 0 and treated with 10 mg/kg SZU-101 for different schedules (i.t. or i.p., continuous or discontinuous), as shown in the experimental protocol. (**C**–**E**) Combination therapy with SZU-101 and JQ-1. Balb/c mice (*n* = 7–8/group) were implanted with 2 × 10^5^ 4T1 cells in both flanks and i.t. treated with SZU-101 and i.p. treated with JQ-1, as shown in the experimental protocol. Tumor volumes (**C**) and tumor weights (**D**) at both injected and uninjected sites were monitored. Survival curves of the mice (**E**) were also recorded. Data represent mean ± SE, * *p* < 0.05, ** *p* < 0.01, *** *p* < 0.001. Tumor growth curves were analyzed by two-way ANOVA with Bonferroni post hoc test. Tumor weights were analyzed by one-way ANOVA with Tukey’s post hoc test. Survival curves were analyzed by log rank test.

**Figure 2 ijms-25-00663-f002:**
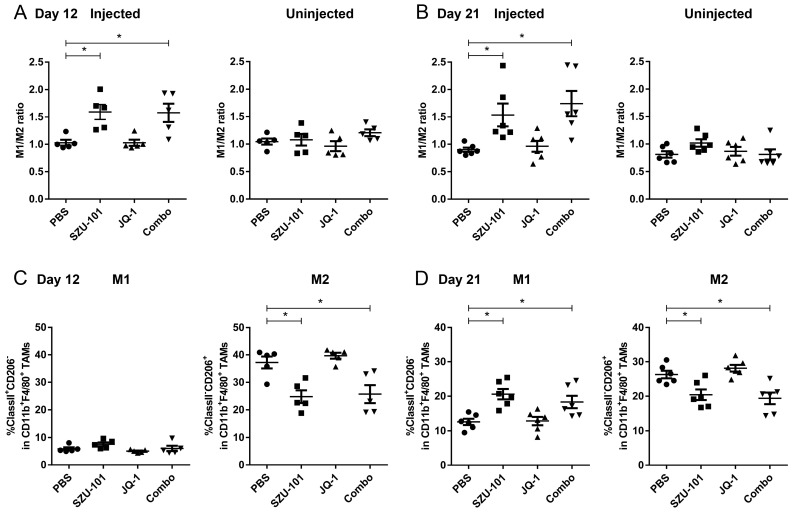
Combination administration of SZU-101 and JQ-1 increased M1/M2 ratio in TAMs. (**A**,**B**) M1/M2 ratio after compound treatment. Balb/c mice (*n* = 5–6/group) were treated with SZU-101 and JQ-1 as described before. Tumors were harvested and tumor-infiltrating cells were analyzed by flow cytometry, and M1/M2 ratios in TAMs of both injected and uninjected sites were monitored on Day 12 (**A**) and 21 (**B**). TAMs were gated on CD45^+^CD11b^+^F4/80^+^ population, and M1/M2 was calculated as the percentage of M1 subset (CD206^−^) divided by M2 subset (CD206^+^) in CD45^+^CD11b^+^F4/80^+^ subset. (**C**,**D**) Kinetics of M1 and M2 population after compound treatment. M1 (MHC^+^CD206^−^ subset) and M2 macrophages (MHC^−^CD206^+^ subset) of the injected site were monitored on Day 12 (**C**) and Day 21 (**D**). Data represent mean ± SE, * *p* < 0.05 (one-way ANOVA with Tukey’s post hoc test).

**Figure 3 ijms-25-00663-f003:**
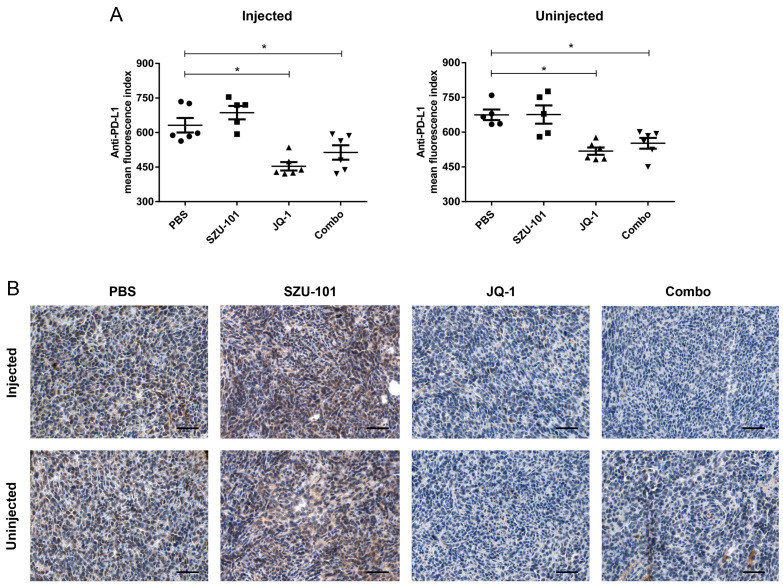
Combination administration of SZU-101 and JQ-1 suppressed PD-L1 expression in tumor cells. (**A**) Balb/c mice (*n* = 5–6/group) were treated with SZU-101 and JQ-1 as described before. PD-L1 levels of tumor cells (CD45^−^ subset) of both injected and uninjected sites were monitored on Day 21. (**B**) Representative images of immunohistochemical staining of tumors on Day 21 for PD-L1 (Scale bars, 50 μm). Data represent mean ± SE, * *p* < 0.05 (one-way ANOVA with Tukey’s post hoc test).

**Figure 4 ijms-25-00663-f004:**
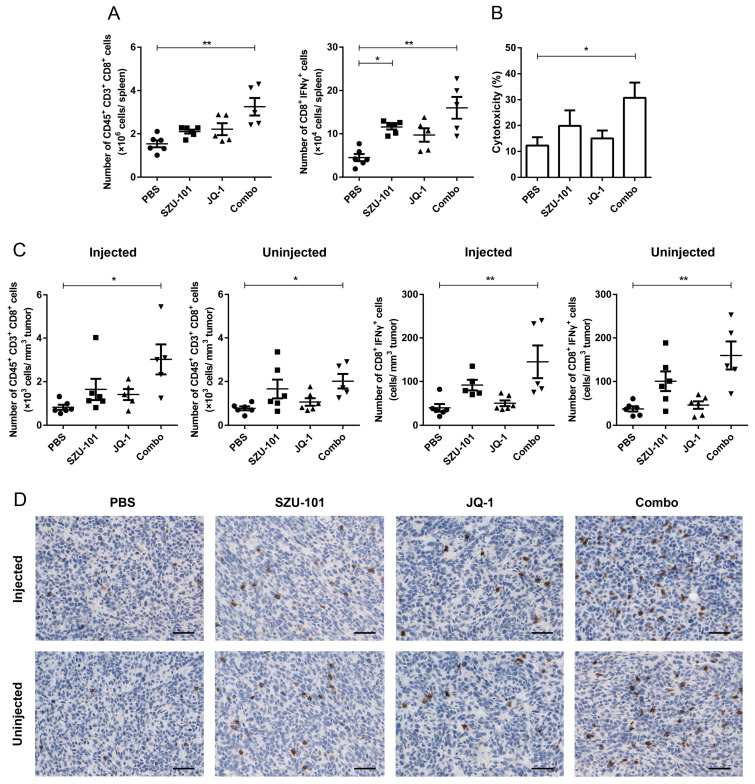
Combination administration of SZU-101 and JQ-1 increased CD8^+^ T cells in spleens and TILs. (**A**) Balb/c mice (*n* = 5–6/group) were treated with SZU-101 and JQ-1 as described before. Spleens were harvested on Day 21, and T cells in spleens were analyzed by flow cytometry. Numbers of CD45^+^CD3^+^CD8^+^ and CD8^+^IFNγ^+^ T cells per spleen were recorded and plotted. (**B**) Cytotoxic T cell responses were determined on Day 21 by incubating spleen lymphocytes (effectors) with 4T1 cells (targets) at the ratio of cell number of 50:1. (**C**) Tumors were harvested on Day 21, and TILs of both injected and uninjected sites were analyzed by flow cytometry. Tumor-infiltrating CD8^+^ T cells were gated on CD45^+^CD3^+^CD8^+^ population, and numbers of CD8^+^ and CD8^+^IFNγ^+^ T cells were recorded and plotted per tumor volume (mm^3^). (**D**) Representative images of immunohistochemical staining of tumors on Day 21 for CD8 (Scale bars, 50 μm). Data represent mean ± SE, * *p* < 0.05, ** *p* < 0.01 (one-way ANOVA with Tukey’s post hoc test).

**Figure 5 ijms-25-00663-f005:**
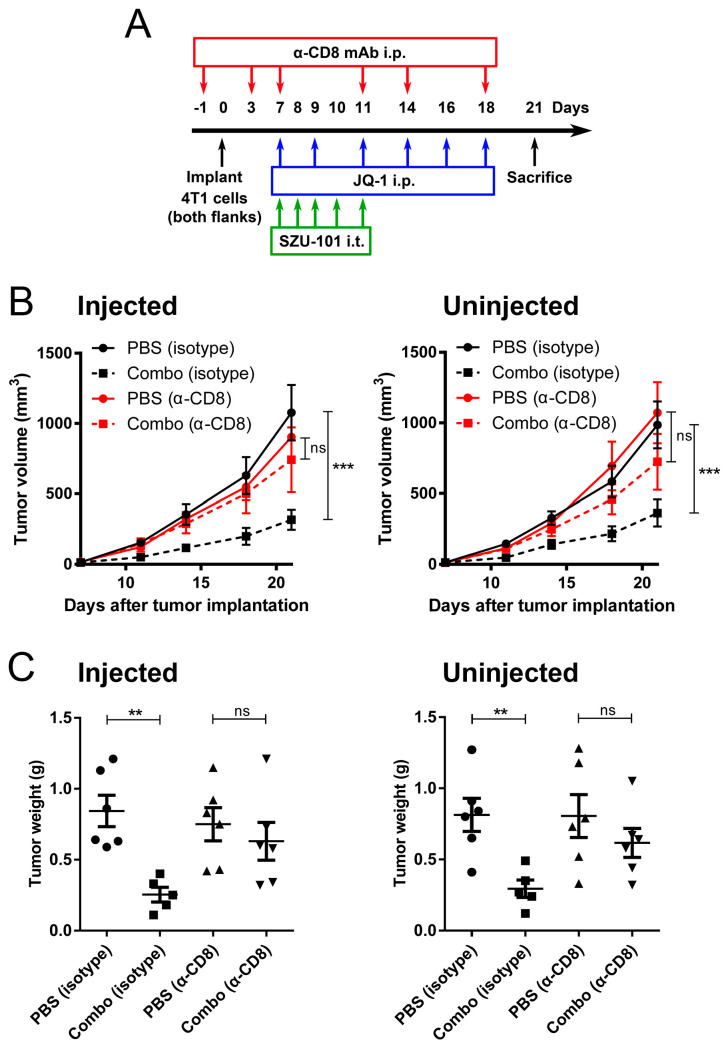
Depletion of CD8^+^ cells abrogated the antitumor effects of combination administration of SZU-101 and JQ-1. (**A**) Experimental protocol of CD8^+^ cell depletion. Balb/c mice (*n* = 5–6/group) were treated with SZU-101 and JQ-1 as described before, and anti-CD8 mAb or isotype control was injected 6 times. (**B**,**C**) Tumor volumes (**B**) and tumor weights (**C**) at both injected and uninjected sites were monitored. Data represent mean ± SE, ** *p* < 0.01, *** *p* < 0.001, ns, statistically non-significant. Tumor growth curves were analyzed by two-way ANOVA with Bonferroni post hoc test. Tumor weights were analyzed by one-way ANOVA with Tukey’s post hoc test.

**Figure 6 ijms-25-00663-f006:**
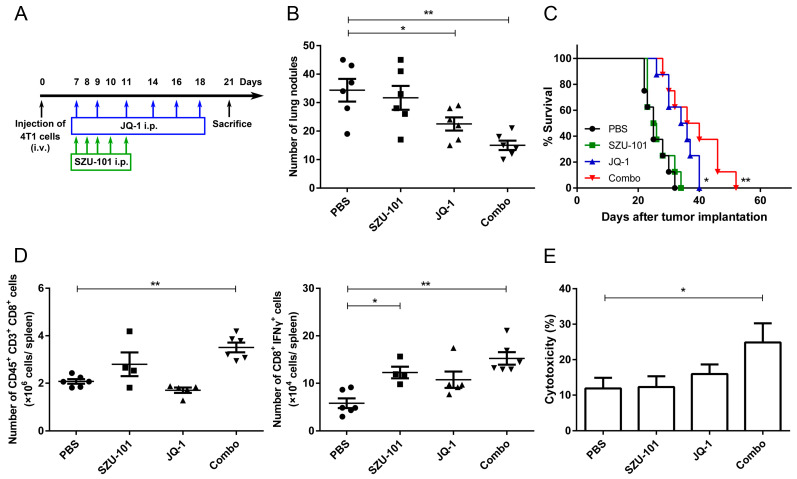
Combination administration of SZU-101 and JQ-1 inhibited tumor metastasis. (**A**) Experimental protocol of tumor lung metastasis. Balb/c mice (*n* = 4–8/group) were intravenously injected through the tail vein with 5 × 10^4^ 4T1 cells on Day 0, and i.p. treated with SZU-101 and JQ-1. (**B**) Numbers of lung nodules were counted on Day 21. (**C**) Survival curves of the mice were also recorded. (**D**) Spleens were harvested on Day 21, and T cells in spleens were analyzed by flow cytometry. Numbers of CD45^+^CD3^+^CD8^+^ and CD8^+^IFNγ^+^ T cells per spleen were recorded and plotted. (**E**) Cytotoxic T cell responses were determined on Day 21 by incubating spleen lymphocytes (effectors) with 4T1 cells (targets) at the ratio of cell number of 50:1. Data represent mean ± SE, * *p* < 0.05, ** *p* < 0.01. Survival curves were analyzed by log rank test. Other experiments were analyzed by one-way ANOVA with Tukey’s post hoc test.

**Figure 7 ijms-25-00663-f007:**
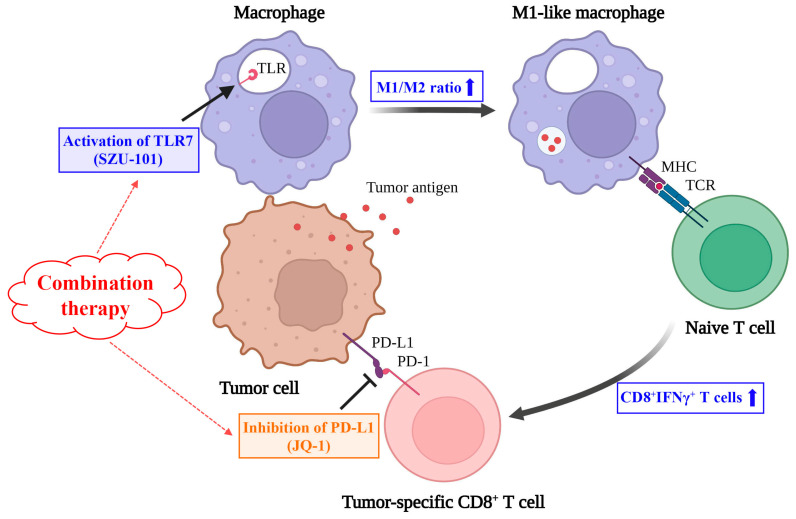
Schematic illustration of combination therapy of SZU-101 and JQ-1 on the suppression of tumor growth.

## Data Availability

The data needed to evaluate the conclusions are present in the paper and/or the Appendix A.

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
