# Peer review of "Combination Therapy with a TLR7 Agonist and a BRD4 Inhibitor Suppresses Tumor Growth via Enhanced Immunomodulation"

_ijms, 2024, doi:10.3390/ijms25010663_

Round 1

Reviewer 1 Report

Comments and Suggestions for Authors

The present work is interesting. However, this publication "https://doi.org/10.1002/ijc.33222" is very similar to this study. The authors must modify their work to avoid this similarity in content and even the idea and animal groups.

- Provide the gross images of tumors from nude mice.

- Test the combination against colony formation, cell migration, and cell invasion assays.

- Perform some bioinformatic analysis to enrich your work.

Author Response

  1. The present work is interesting. However, this publication "https://doi.org/10.1002/ijc.33222" is very similar to this study. The authors must modify their work to avoid this similarity in content and even the idea and animal groups.

Thanks for the comments. We reported the antitumor effects of a chemical conjugation of SZU-101 and JQ-1 (SZU-119) in the publication https://doi.org/10.1002/ijc.33222, where JQ-1 alone was included as a comparison group. In that publication, JQ-1 should be given to the mice with the same dose and schedule as SZU-119 (10 mg/kg, continuous i.t. treatment for 5 times). However, JQ-1 alone could achieve stronger antitumor effects without apparent toxicity, by being administrated with a higher dose (50 mg/kg) and a discontinuous i.p. treatment for 5 times. Therefore, the novelty of this work was to provide a strategy of combination therapy with SZU-101 and JQ-1 with different drug administration plans, which exerted more potent effects on tumor growth than the chemical conjugation SZU-119. The above explanation was added to the section of “Discussion”.

  1. Provide the gross images of tumors from nude mice.

Thanks for the comments. We added the representative images of tumors and lung nodules from mice, as displayed in Fig. S4 and Fig. S8.

  1. Test the combination against colony formation, cell migration, and cell invasion assays.

Thanks for the comments. The hypothesis of this study was that combination therapy of SZU-101 and JQ-1 suppressed tumor growth by the regulation of macrophages and CD8+ T cells in the TME. In other words, the antitumor effects of combination therapy were exerted indirectly via immune cells, rather than directly via tumor cells. Thus, the effects of colony formation, cell migration and cell invasion of the combination would be the same as those of JQ-1, where SZU-101 was supposed to be inactive to tumor cells in vitro. Therefore, we did not test these bioactivities of the combination, and thanks for your understanding.

  1. Perform some bioinformatic analysis to enrich your work.

Thanks for the comments. As we mentioned above, in our hypothesis, the antitumor effects of combination therapy were exerted indirectly via immune cells, rather than directly via tumor cells. Thus, the signaling pathways in tumor cells of the combination therapy were supposed to be same as SZU-101 alone plus JQ-1 alone, which were well-defined before as TLR7 pathways and BRD4 pathways. Therefore, we did not perform the bioinformatic analysis, and thanks for your understanding.

Reviewer 2 Report

Comments and Suggestions for Authors

Journal: International Journal of Molecular Sciences

Manuscript number#: IJMS-2768191

Article type: Research Paper

Review Decision: Major Revision / Rejection

Title: Combination therapy with a TLR7 agonist and a BRD4 inhibitor suppresses tumor growth via enhanced immunomodulation

Review Report:

This study provides interesting preliminary evidence for the potential anticancer effects of TLR7 agonist SZU-101 and BRD4 inhibitor JQ-1. However, there are a few drawbacks of the study that need to be addressed before accepting the manuscript for publication.

Comments:

·         Future clinical studies are mentioned in the abstract; however, a more thorough examination of the possible obstacles and factors to be taken into account for clinical translation is required.

·         In the introduction, provide more information on previous research that has been carried out to make a strong background for the rationale of the study.

·         The specific methods by which SZU-101 and JQ-1 synergistically boost CD8+ T cell responses and decrease tumor development could be further investigated, even though the immunomodulatory effects are evident. Signaling pathways or transcriptional changes could be investigated and mentioned in this study.

·         The study mainly focuses on two murine models (breast cancer and melanoma). Testing the combination therapy in additional cancer types would broaden its generalizability. Moreover, the authors explain the rationale for choosing only these murine models.

·         While the downstream effects are well-described, a deeper exploration of the molecular mechanisms by which the combination therapy exerts its immunomodulatory effects would be valuable.

·         RNA and ChIP seq techniques should be used by the authors to identify the gene expression changes along with the proteins or transcription factors interacting with SZU-101 and JQ-1 target genes.

·         More reference studies should be cited for the results of this study to make a connection between your findings and previous research. A comparison should be drawn between the results of current and previous studies.

·         Redraw Figure 7 to explain the conclusion drawn from your study. This basic illustration does not explain what’s happening without reading the whole manuscript. A Schematic illustration should be self-explanatory with adequate explanation.

Comments on the Quality of English Language

Minor editing of English language required

Author Response

  1. In the introduction, provide more information on previous research that has been carried out to make a strong background for the rationale of the study.

Thanks for the comments. We cited and explained some previous research with similar rationale of our study, like that a TLR7 agonist or a TLR9 agonist (i.t. treatment) enhanced the efficacy of anti-PD-1 antibody (i.p. treatment) through the activation of CD8+ T cells, by simultaneously targeting innate immune cells and adaptive immune cells. The above explanation was added to the section of “Introduction”.

  1. The specific methods by which SZU-101 and JQ-1 synergistically boost CD8+ T cell responses and decrease tumor development could be further investigated, even though the immunomodulatory effects are evident. Signaling pathways or transcriptional changes could be investigated and mentioned in this study.

Thanks for the comments. In this paper, we displayed that CD8+ T cell responses were related to the increase of M1/M2 ratio and the decrease of PD-L1 expression, and we agreed that immunomodulatory effects could be explained in more detail, such as the ability of antigen uptake of macrophages, and the frequency of TCR clones of T cells. The above explanation was added to the section of “Discussion”. However, we did not further perform experiments due to the limited time of revision, and thanks for your understanding.

  1. The study mainly focuses on two murine models (breast cancer and melanoma). Testing the combination therapy in additional cancer types would broaden its generalizability. Moreover, the authors explain the rationale for choosing only these murine models.

Thanks for the comments. We agreed that testing the combination therapy in other types of cancers would broaden its generalizability, and the above explanation was added to the section of “Discussion”. Moreover, we chose 4T1 breast cancer model and B16 melanoma model, because the antitumor effects of TLR7 agonists or BRD4 inhibitors were decided previously in these two types of cancers. The above explanation was added to the section of “Introduction”.

  1. While the downstream effects are well-described, a deeper exploration of the molecular mechanisms by which the combination therapy exerts its immunomodulatory effects would be valuable. RNA and ChIP seq techniques should be used by the authors to identify the gene expression changes along with the proteins or transcription factors interacting with SZU-101 and JQ-1 target genes.

Thanks for the comments. The hypothesis of this study was that combination therapy of SZU-101 and JQ-1 suppressed tumor growth by the regulation of macrophages and CD8+ T cells in the TME. In other words, the antitumor effects of combination therapy were exerted indirectly via immune cells, rather than directly via tumor cells. Thus, the molecular mechanisms in tumor cells of the combination therapy were supposed to be same as SZU-101 alone plus JQ-1 alone, which were well-defined before as TLR7 pathways and BRD4 pathways. Therefore, we did not perform the RNA and ChIP analysis, and thanks for your understanding.

  1. More reference studies should be cited for the results of this study to make a connection between your findings and previous research. A comparison should be drawn between the results of current and previous studies.

Thanks for the comments. We reported the antitumor effects of a chemical conjugation of SZU-101 and JQ-1 (SZU-119) (Int J Cancer, 2021, 148(2): 437-447), where JQ-1 alone was included as a comparison group. In that publication, JQ-1 should be given to the mice with the same dose and schedule as SZU-119 (10 mg/kg, continuous i.t. treatment for 5 times). However, JQ-1 alone could achieve stronger antitumor effects without apparent toxicity, by being administrated with a higher dose (50 mg/kg) and a discontinuous i.p. treatment for 5 times. Therefore, the novelty of this work was to provide a strategy of combination therapy with SZU-101 and JQ-1 with different drug administration plans, which exerted more potent effects on tumor growth than the chemical conjugation SZU-119. The above explanation was added to the section of “Discussion”.

  1. Redraw Figure 7 to explain the conclusion drawn from your study. This basic illustration does not explain what’s happening without reading the whole manuscript. A Schematic illustration should be self-explanatory with adequate explanation.

Thanks for the comments. We revised the schematic illustration with more details, as displayed in Figure 7.

Round 2

Reviewer 1 Report

Comments and Suggestions for Authors

The authors respond to the comments.

Reviewer 2 Report

Comments and Suggestions for Authors

Journal: International Journal of Molecular Sciences

Manuscript number#: IJMS-2768191

Article type: Research Paper

Review Decision: Accepted

Title: Combination therapy with a TLR7 agonist and a BRD4 inhibitor suppresses tumor growth via enhanced immunomodulation

Review Report:

This manuscript has been substantially improved in clarity, detail, and scientific rigor as a result of the authors' comprehensive responses to my concerns. Authors have provided valuable insight into the synergistic effects of SZU-101 and JQ-1 in suppressing tumor growth in their revised work.

As a result of including relevant previous research in the introduction, the rationale for the study has been strengthened. It is understandable that the authors did not conduct a deeper mechanistic investigation owing to time constraints. Additionally, the rationale for choosing specific murine models is adequately explained.

The revised discussion section effectively expands on the immunomodulatory effects and broader applicability of the findings. The focus of the authors' work on immune cell regulation is justified by their explanation of why they did not do RNA and ChIP seq analysis. Furthermore, the additional references effectively connect the current findings with previous research. Finally, the revised Figure 7 clearly illustrates the main conclusions drawn from the study.

While some minor limitations may remain, they are overshadowed by the significant improvements made. Encouraging the authors to pursue further investigations into the molecular mechanisms in future studies would be beneficial. Considering the significant advancement that has been shown, I advise accepting the updated paper for publication.